# Assessment of Eating Disorder Risk According to Sport Level, Sex, and Social Media Use among Polish Football Players: A Cross-Sectional Study

**DOI:** 10.3390/nu16203470

**Published:** 2024-10-14

**Authors:** Wiktoria Staśkiewicz-Bartecka, Grzegorz Zydek, Małgorzata Magdalena Michalczyk, Marek Kardas, Oskar Kowalski

**Affiliations:** 1Department of Food Technology and Quality Evaluation, Faculty of Public Health in Bytom, Medical University of Silesia in Katowice, 41-808 Zabrze, Poland; mkardas@sum.edu.pl; 2Department of Sport Nutrition, Jerzy Kukuczka Academy of Physical Education in Katowice, ul. Mikołowska 72A, 40-065 Katowice, Poland; g.zydek@awf.katowice.pl; 3Institute of Sport Sciences, Jerzy Kukuczka Academy of Physical Education in Katowice, ul. Mikołowska 72A, 40-065 Katowice, Poland; m.michalczyk@awf.katowice.pl; 4Department of Dietetics and Human Nutrition, Faculty of Public Health in Bytom, Medical University of Silesia in Katowice, 41-808 Zabrze, Poland; okowalski@sum.edu.pl

**Keywords:** eating disorders, football, soccer, sport level, sex, social media, mental health

## Abstract

Background/Objectives: Eating disorders (EDs) pose a significant health issue affecting athletes, with risk factors varying by sport level, sex, and social media use. This study assesses the risk of EDs among professional and amateur football players, considering these factors, and compares findings with a control group of non-athletes. Methods: The study involved 170 participants, including non-athletes as a control group, categorized by sex and sport level. The mean age of participants was 24.3 ± 4.20, with an age range of 18–36. The Eating Attitudes Test (EAT-26) and body mass index (BMI) assessments were used to determine ED risk. Results: Results showed a higher prevalence of ED risk among professional athletes, especially women, compared to amateurs and non-athletes. Social media use and body comparisons were linked to increased ED risk, with professional athletes exhibiting higher vulnerability due to performance pressures. Women, particularly those in professional sports, showed a higher risk of EDs than men, influenced by social and aesthetic pressures. Conclusions: The findings highlight the need for targeted interventions, promoting healthier body image perceptions and addressing social media’s role in shaping body dissatisfaction. Psychological support and sex-specific strategies should be integrated into athlete care programs to mitigate these risks.

## 1. Introduction

Eating disorders (ED) are a significant health issue affecting millions of people worldwide. According to the World Health Organization (WHO), eating disorders, such as anorexia nervosa (AN), bulimia nervosa (BN), and binge eating disorder (BED), have not only severe health consequences but also psychological ones, leading to long-term problems in social and occupational functioning [1,2]. Traditionally, these disorders were primarily associated with young women and sports disciplines where body shape plays a crucial role, such as gymnastics, ballet, or athletics [3]. However, an increasing number of studies show that this issue also affects team sports athletes, including male and female football players [4].

EDs are categorized based on criteria established in two predominant systems for classifying mental illnesses. In Europe, the ICD-11 (International Classification of Diseases) is utilized, whereas in the United States, the DSM-5 (Diagnostic and Statistical Manual of Mental Disorders) is the standard reference [5,6]. Both systems offer comprehensive descriptions of various EDs, including AN, BN, and other related conditions, facilitating precise diagnosis and the development of suitable treatment plans for those affected [5,6]. The DSM-5 and ICD-11 serve as crucial diagnostic resources for mental health practitioners, ensuring accurate identification and appropriate management of EDs. Additionally, they emphasize the necessity for continued research into the underlying mechanisms and therapeutic approaches for these disorders to enhance the support available to those in need [7,8].

Football, as one of the most popular sports in the world, plays a key role in shaping both the bodies and minds of athletes. Competition at a high level, pressure to achieve, and the need to maintain a certain body shape can be significant risk factors for the development of EDs [9]. In particular, professional athletes are often subjected to rigorous standards regarding body weight and composition, which can lead to unhealthy eating habits [10,11]. Research also indicates differences in the risk of developing EDs between professional and amateur athletes, making this group particularly interesting to study [12].

Additionally, in recent years, the growing influence of social media has played an important role in shaping body perception, especially among young people and athletes [13]. Platforms such as Instagram, TikTok, and Facebook frequently promote unrealistic beauty standards, which can affect users’ well-being and eating habits. Studies show that individuals who regularly use social media may be more at risk of developing EDs, as they are exposed to body comparisons, the promotion of idealized physiques, and pressure to meet socially accepted beauty standards [14,15].

Considering both sex and level of involvement in sports, there are significant differences in how male and female athletes respond to these factors [16]. Men, although less prone to EDs than women, may also experience body-image-related issues, especially in sports that promote a particular body shape [17]. Women, on the other hand, are more susceptible to the influence of social media and aesthetic pressures, making sex-based comparative studies on ED risk particularly important [18].

The purpose of this study is to examine the prevalence of ED risk among Polish male and female football players. Furthermore, the study seeks to analyze the variations in this risk based on sport level, sex, and the extent of social media usage.

In preparation for the study, the following research hypotheses were established: There is a significant association between the risk of developing EDs and variables such as sport level (professional vs. amateur), sex, and the frequency of social media use among male and female football players. It is expected that professional athletes, especially female athletes, and those with higher social media exposure, will exhibit a higher risk of EDs compared to amateur athletes and individuals with lower social media usage. Additionally, the control group (non-athletes) will show a lower prevalence of ED risk than both professional and amateur athletes.

## 2. Materials and Methods

### 2.1. Procedure for the Study

The study was conducted between April and August 2024 using a mixed method with a survey technique. Professional male and female football players participated in the direct part of the study. The questionnaires were completed at the club after training, and before filling them out, participants received instructions to ensure proper understanding and interpretation of the questions and to minimize errors in their responses.

Simultaneously, an indirect study (CAWI method) was conducted among amateur male and female football players, as well as a control group of women and men who were not physically active. Participants received QR codes directing them to the survey questionnaire, which is an acceptable method in psychological research. Before providing the QR codes, the researchers gave detailed instructions to ensure proper understanding of the questions and to minimize the possibility of errors when completing the survey. Respondents completed the questionnaires on the day they received the QR codes. Google Forms was used to collect the data due to its ease of use, accessibility, and the ability to customize the questionnaire to the specific needs of the study. The average time required to complete the survey was approximately 20 min.

The study sample was deliberately chosen to reflect the characteristics, unique experiences, and attributes pertinent to the research topic, which was essential for fulfilling the study’s aims.

All participants in the study were thoroughly informed about its purpose and assured of their anonymity, with their consent obtained for data usage. Details about the voluntary and informed nature of their participation were provided at the start of the survey. This study was conducted in compliance with the Declaration of Helsinki, as set out by the World Medical Association. Ethical approval was granted by the Bioethics Committee of the Medical University of Silesia in Katowice (BNW/NWN/0043-3/641/35/23, approval date: 22 September 2023), in accordance with the Act of 5 December 1996, concerning the professions of physicians and dentists (Journal of Laws 2016, item 727).

### 2.2. Participants

A total of 170 respondents participated in the study, all of whom were of Polish nationality. The mean age of participants was 24.3 ± 4.20, with an age range of 18–36. The professional male footballers (PFM) who took part in the study were players from two clubs competing in the Ekstraklasa—the highest men’s football league in Poland (n = 23). The professional female footballers (PFW) represented a club playing in the Ekstraliga—the highest level of women’s football in Poland (n = 26). Meanwhile, the amateur male footballers (AFM) belonged to a club competing in the IV men’s league (n = 26), and the amateur female footballers (AFW) played in a club at the IV women’s league level (n = 30), in accordance with PZPN regulations [19]. The control group consisted of 34 women (CGW) and 31 men (CGM) who were not physically active. To ensure reliable comparison of the results, the following inclusion criteria were established for the control group: (1) aged 18–36 years, (2) no participation in physical activity more than once a week, lasting at least 60 min, (3) no past or present eating disorders diagnosed by a doctor, (4) consent to participate in the study, and (5) complete filling out of the study questionnaire.

### 2.3. Survey Tools

The study was conducted using a survey questionnaire, which included a demographic section (collecting information such as age, height, weight, chronic illnesses and medications, education, sources of nutritional knowledge, and food exclusions) and a validated tool for assessing the risk of EDs: the Eating Attitudes Test (EAT-26). The EAT-26 questionnaire includes questions about the lowest, highest, and subjectively perceived ideal body weight in adult life. Additionally, participants were asked about their average daily time spent on social media, the most common types of applications used, and whether they compare their body image to photos on social media. Questions also addressed their satisfaction with their body appearance, providing a comprehensive view of how social media influences body image and self-perception among athletes.

#### 2.3.1. Body Mass Index (BMI)

The participants’ nutritional status was evaluated using the body mass index (BMI), calculated with the following formula:$$BMI=\frac{body\;weight\;(kg)}{{height\;(m)}^{2}}$$

The results were interpreted based on the guidelines provided by the WHO [20].

#### 2.3.2. EAT-26

The study utilized the American Eating Attitudes Test (EAT-26) developed by Garner et al. [21] as a screening tool for evaluating the risk of EDs. This questionnaire is a standardized instrument designed to identify symptoms indicating a risk for EDs. It was specifically created to assess both individuals with clinical diagnoses and those at risk for AN, BN, or obesity. The EAT-26 is among the most widely recognized diagnostic tools used in global studies on the prevalence of EDs. The Polish version of the tool was standardized by K. Włodarczyk-Bisaga [22]. Interpretation of the EAT-26 is based on three “referral criteria” that determine whether the participant should undergo further evaluation for EDs risk:The final score on the EAT-26 is calculated by summing the responses to 26 questions that assess attitudes toward food. Responses to questions 1 through 25 are scored as follows: always = 3 points, usually = 2 points, often = 1 point, and other answers = 0 points. Question 26 is scored in reverse, with never = 3 points, and so on. The total score ranges from 0 to 78, with a score of 20 or higher indicating a risk of developing an ED and the need to consult a specialist for further evaluation.Questions related to behaviors may indicate symptoms of an ED or recent significant weight loss. These questions assess compensatory behaviors, such as using laxatives, inducing vomiting, binge eating, excessive exercise, or rapid, substantial weight loss over a short period. An affirmative response to any of these questions suggests the possibility of abnormalities and the need for further investigation of ED risk.The questionnaire also includes specific questions regarding height, weight, and sex, which are used to calculate BMI. If the calculated BMI is below the age-appropriate standard, it may indicate a potential risk for an ED. Assessing BMI in relation to the participant’s height, weight, and sex helps to identify potential risks and the need for further evaluation. Table 1 presents BMI interpretations in comparison with age-related standards.

### 2.4. Statistical Analysis

Statistical analyses were performed using Statistica v.13.3 (Stat Soft Poland) and the R package v. 4.0.0 (2020) under the GNU GPL (The R Foundation for Statistical Computing). To present quantitative data, mean values and standard deviations (X ± S) were calculated; for qualitative data, percentage notation was used.

The normality of the distribution was assessed using the Shapiro–Wilk test. The Chi2 tests of independence or Fisher’s exact tests were applied to assess the associations between categorical variables, such as sex, education level, and sources of nutritional knowledge. To evaluate differences in BMI between groups, one-way ANOVA tests were conducted. An ANCOVA was conducted to investigate significant interactions between the variables, i.e., sport level and sex. The Chi2 tests of independence or Fisher’s exact tests were also used to investigate the relationships between ED risk and variables, such as time spent on social media, the social media platforms used, body image comparisons, and body satisfaction.

Pairwise comparisons of body weight measurements were performed using the Friedman test to determine differences between measurements taken in the same group and the Durbin–Conover test.

Cramer’s V coefficient was employed to explore the relationship between EAT-26 interpretation and variables such as sex and athletic level. This coefficient quantifies the strength of the association between two categorical variables, in this case, the risk score for developing an ED.

A linear regression analysis was conducted to assess the relationship between various dimensions of body mass and the EAT-26 score, which measures the risk of ED. The results of the analysis are presented in the form of regression coefficients (estimates), along with standard deviations, t-statistics, and statistical significance levels.

A value of *p* < 0.05 was used as a criterion for statistical significance.

## 3. Results

One hundred and seventy people participated in the study, and the participants were divided into six groups according to their sports level and sex. The first group consisted of male professional football players (n = 23; PFM), and the second group consisted of female professional football players (n = 26; PFW). Another two groups consisted of male amateur football players (n = 26; AFM) and female amateur football players (n = 30; AFW). The two control groups were physically inactive women (n = 34; CGW) and physically inactive men (n = 31; CGM). A total of 80 men (47.1%) and 90 women (52.9%) participated in the study, including 49 (28.8%) professional athletes, 56 (32.9%) amateurs, and 65 (38.2%) physically inactive individuals.

The majority of respondents had secondary education (n = 108; 63.5%) and higher education (n = 56; 32.9%). Statistically significant differences were found between sex and education level (*p* = 0.012), with women more often having higher education than men (43.3% and 21.3%, respectively). Statistically significant differences were also found between sports level and education (*p* = 0.041), with the highest percentage of individuals with higher education in the control group (n = 26; 40%), followed by amateurs (n = 19; 33.9%) and professional athletes (n = 11; 22.4%). Five individuals (four women and one man) suffered from chronic illnesses: two had asthma, and three had allergies. The characteristics of the study group are presented in Table 2.

The study results indicated no statistically significant differences in sources of nutritional knowledge between sexes (*p* = 0.345), but revealed significant differences based on sports level (*p* < 0.01). Women more often than men used the services of a dietitian (21.1% compared to 23.8%) and more frequently obtained knowledge from the internet (55.6% vs. 42.5%). Men from the control group were more likely to not acquire any nutritional knowledge (41.9% compared to 20.6% in the women’s group). Among professional athletes, the most common source of knowledge was a dietitian, with 69.6% of men and 15.4% of women using this service. In contrast, in the amateur groups, the main source of knowledge was the internet, used by 61.5% of male amateurs and 56.7% of female amateurs. Detailed information is presented in Table 3. There were no significant correlations between food exclusions from the diet and athletic level, only women in the control group were statistically significantly more likely to exclude red meat from their diet (*p* < 0.01).

Based on the presented results, most respondents did not exclude specific food groups from their diet. In the entire studied population, 68% of individuals did not exclude any food groups from their daily diet, while 32% reported such exclusions. Significant differences were found in the exclusion of red meat (*p* = 0.006) between the study groups. Only four individuals (2%) from the entire population declared avoiding red meat, all of whom were from the control group of women, where 12% of respondents excluded this product. No other group reported red meat exclusion, suggesting that women outside the sports environment are more likely to avoid this product. Similarly, for gluten-containing grains (*p* = 0.045), significant differences were observed between the groups. Exclusion of these products was more common among amateur female football players and professional male football players. In the group of amateur female football players, 13% excluded gluten, while in the group of professional male football players, it was 9%. In the remaining cases (soy, lactose, dairy, fish, eggs, meat, and simple sugars), no statistically significant differences were found between the studied groups.

The participants in the study were examined regarding their social media activity, comparisons of their body shape with other images on social media, and their body satisfaction. Detailed information can be found in Table 4.

### 3.1. BMI of Participants

According to the interpretation of the BMI, it was shown that most of the subjects, 78.8% (n = 134), had a BMI within the normal range. At the same time, 14.1% (n = 24) of the subjects were classified as overweight, which is a significant portion of the population. Underweight was present in 4.7% (n = 8) of the participants. Grade I obesity was found in 1.8% (n = 3) of participants, while grade III obesity occurred in only one person (0.6%). Statistically significant differences were found between sex (*p* < 0.001) and sports level (*p* < 0.001) and nutritional status. Detailed information on nutritional status is shown in Figure 1.

The BMI for the entire study group (n = 170) averaged 22.7 ± 3.26 kg/m^2^. Among men (n = 80), the mean BMI was higher at 24.2 ± 2.66 kg/m^2^, while among women (n = 90), the mean BMI was lower, averaging 21.4 ± 3.22 kg/m^2^. No statistically significant differences were found between sports level and BMI in either sex group (*p* = 0.050 for men and *p* = 0.056 for women). Detailed information is presented in Figure 2.

The ANCOVA revealed significant differences in BMI based on sports level (*p* = 0.006) and sex (*p* < 0.001). This indicates that both sports level and sex had a significant impact on BMI values. The interaction between sports level and sex was not statistically significant (*p* = 0.635), suggesting that the effect of sports level on BMI was similar for both sexes.

### 3.2. Analysis of Body Mass Changes and Subjective Assessment of Ideal Body Mass

Table 5 compares body mass among the study groups regarding current, lowest, highest, and ideal body mass. Among men, the current body mass was 77.7 ± 9.65 kg, and the differences between their current, highest, and lowest weights were statistically significant (*p* < 0.001). However, there was no significant difference between the current and ideal mass (*p* = 0.322). In contrast, for women, whose current body mass was 59.4 ± 9.31 kg, these differences were significant in all comparisons (*p* < 0.001), indicating a greater discrepancy between the current body mass and the perceived ideal body mass. In the individual subgroups based on athletic level, the results did not show variability in current (*p* = 0.390), lowest (*p* = 0.899), highest (*p* = 0.128), or ideal (*p* = 0.649) body mass. However, differences were found between men and women in current, lowest, highest, and ideal body mass (*p* < 0.01; Table 5).

### 3.3. Risk of ED

Based on the EAT-26 results from Table 6, it was found that 10.0% of all respondents (n = 170) were identified as being at elevated risk for EDs in Part A, while 16.5% demonstrated elevated risk in Part B, and 9.4% in Part C. Overall, 33.5% of the participants met at least one criterion suggesting a potential risk of ED. Significant differences were found between males and females, with females showing a higher risk in Part A (15.6% vs. 3.8%, *p* = 0.010) and in the overall score (43.3% vs. 22.5%, *p* = 0.004). Additionally, differences were observed when comparing professional athletes, amateurs, and the control group (*p* = 0.002 for Part A, *p* < 0.001 for Part B, and *p* < 0.001 for the overall result (Table 6)).

Statistically significant differences were found between ED risk and nutritional status, as assessed by BMI interpretation (*p* < 0.001). The majority of participants with a normal BMI (n = 93; 69.4%) were not at risk for ED, while 30.6% (n = 41) of those with a normal BMI displayed a potential risk. Among participants classified as overweight, 70.8% (n = 17) were not at risk, while 29.2% (n = 7) were identified as being at risk. All participants classified as underweight (n = 8; 100%) were considered to be at risk of ED, highlighting the heightened vulnerability of this group. In contrast, none of the individuals with first-degree obesity were at risk, while the single participant classified with third-degree obesity was identified as being at risk. There were no statistically significant differences between ED risk and age (*p* = 0.222) or education (*p* = 0.734).

The results of the analysis did not indicate statistically significant differences between sex and the time spent on social media in relation to ED risk (*p* = 0.325). For men, the association between time spent on social media and ED risk was not significant (*p* = 0.152). Similarly, for women, there was no significant association (*p* = 0.159). Significant differences were observed in the use of social media platforms and ED risk, particularly in relation to the use of Twitter and YouTube in amateur football players. For players, those who frequently used Twitter and YouTube were more likely to be at risk for EDs (*p* = 0.006), with a moderate effect size indicated by a Cramer’s V value of 0.505. The results of the analysis indicated statistically significant differences between sex and the tendency to compare body shape with images of other athletes on social media. For men, a significant association was found between comparison behavior and ED risk (*p* < 0.001), where those who frequently compared their body shape were more likely to be at risk for EDs. The Cramer’s V value of 0.493 suggested a moderate effect size. In contrast, for women, no significant association was found between comparison behavior and ED risk (*p* = 0.423). However, when considering the entire study group, a significant relationship was observed (*p* = 0.002), with a Cramer’s V value of 0.270, indicating a moderate effect size. The analysis revealed statistically significant differences in body satisfaction and ED risk for specific groups. For the overall study group, those who were not satisfied with their body and would like to change many things were more likely to be at risk for EDs (*p* = 0.03), with a moderate effect size indicated by a Cramer’s V value of 0.204. Those fully satisfied with their appearance were significantly less likely to be at risk for EDs (*p* = 0.031), with a moderate effect size (Cramer’s V = 0.278). When considering subgroups by sports level, dissatisfaction with body appearance among professional football players (PF) was significantly associated with an increased ED risk (*p* = 0.031), with a moderate effect size (Cramer’s V = 0.278). Additionally, for amateur football players, dissatisfaction was significantly related to ED risk (*p* = 0.013), with a larger effect size (Cramer’s V = 0.393). Detailed results are shown in Table 7.

#### 3.3.1. Linear Regression Analysis of EAT-26 and Current, Lowest, Highest, and Ideal Body Weight by Gender

Table 8 presents the results of a linear regression analysis aimed at examining the relationship between various predictors, such as body mass, highest body mass in adulthood, lowest body mass in adulthood, ideal body mass, and gender, with the dependent variable being the total score obtained in the EAT-26 test. The model’s fit measure (R^2^ = 0.0565) suggested that approximately 5.65% of the variability in the total score obtained in the EAT-26 test could be explained by these predictors, indicating a weak overall fit.

The analysis showed that the intercept was not statistically significant (*p* = 0.567), meaning there was no evidence that the baseline score (without any predictors) differed significantly from zero. None of the predictors related to body mass—current mass (*p* = 0.245), highest mass (*p* = 0.455), lowest mass (*p* = 0.381), or ideal body mass (*p* = 0.097)—had statistically significant effects on the total score obtained in the EAT-26 test. A significant effect was found for gender (*p* = 0.005), with men scoring, on average, 5.40 points lower than women, indicating a gender-based difference in the total score obtained in the EAT-26 test. Overall, the model highlighted the limited role of body-mass-related predictors, while gender showed a more pronounced and significant effect.

#### 3.3.2. Linear Regression Analysis of EAT-26 and Current, Lowest, Highest, and Ideal Body Weight by Sports Level

Table 9 presents the results of a linear regression analysis aimed at examining the relationship between various predictors, such as body mass, highest body mass in adulthood, lowest body mass in adulthood, ideal body mass, and group membership (PF, AF, and CG), with the dependent variable being the total score obtained in the EAT-26 test. The model’s fit measure (R^2^ = 0.0352) suggested that approximately 3.52% of the variability in the total score obtained in the EAT-26 test could be explained by these predictors, indicating a weak overall fit.

The analysis showed that the intercept was statistically significant (*p* < 0.001), meaning that the baseline score differed significantly from zero. However, none of the predictors related to body mass—current mass (*p* = 0.596), highest mass (*p* = 0.181), lowest mass (*p* = 0.190), or ideal body mass (*p* = 0.714)—had statistically significant effects on the total score obtained in the EAT-26 test. Regarding group membership, the comparison between AF and PF did not show a significant effect (*p* = 0.509), while the comparison between CG and PF was statistically significant (*p* = 0.049), indicating that the CG scored, on average, 2.75 points lower than the PF group.

Overall, the model highlighted the limited role of body-mass-related predictors in explaining the variability in the total score obtained in the EAT-26 test, while some differences between groups (specifically CG versus PF) seemed to be statistically significant.

## 4. Discussion

This study aimed to assess the risk of EDs among professional and amateur football players, as well as a non-sporting control group, taking into account the role of social media, sports level, and sex. The findings confirmed that EDs are a significant issue among athletes, with factors such as sex and sports level playing a crucial role in the risk of developing these disorders. Furthermore, the study highlighted the growing influence of social media on body perception, particularly among female athletes.

The results showed that both professional and amateur athletes were more prone to developing EDs compared to the non-sporting control group. Among the athletes, professional female football players demonstrated the highest risk of EDs—42.3% of this group showed an elevated risk based on the EAT-26 test results. These findings are consistent with previous research, which emphasizes that women, especially those in sports requiring a specific body shape, are more susceptible to EDs [23]. Interestingly, male professional football players also showed an elevated risk (43.5%) in Part B of the EAT-26 test, which assesses behaviors such as binge eating, vomiting, and excessive physical activity. This demonstrates that male athletes in professional sports also face pressure to maintain appropriate body weight and physical condition.

On the other hand, male amateur football players exhibited a relatively lower risk (19.2%) of developing EDs, suggesting that the pressure of competition and body appearance expectations in professional sports are stronger risk factors than amateur-level sports participation. This trend was also observed in the control group, where the prevalence of ED risk was significantly lower, indicating that participation in elite sports, rather than the general absence of physical activity, is a key factor influencing the risk of EDs [24]. The significant differences between the control group and professional athletes suggest that professional athletes may be at a higher risk of EDs compared to non-athletes. This could be due to the pressure of achieving and maintaining a specific body shape to meet the requirements of their sport, which in turn may lead to weight control behaviors and restrictive diets.

The study revealed significant sex differences in the risk of EDs, with a higher prevalence of disorders in women than in men. Overall, 43.3% of women met at least one criterion for the risk of EDs, compared to 22.5% of men. This finding corroborates previous studies that suggest women are more vulnerable to EDs due to societal and aesthetic pressures [25,26,27]. The significant difference between the scores of women and men, with women showing higher scores on the EAT-26 scale, may suggest that women are more susceptible to developing EDs. This could be due to greater societal pressure regarding appearance and adherence to certain beauty standards, which women face more often than men. This pressure is often exacerbated by social media and the “ideal body” culture, which can lead to negative body image and unhealthy eating behaviors.

The study results indicated that women in the control group were significantly more likely to compare their bodies to images on social media (52.9%), suggesting that women outside the sports environment are more susceptible to the influence of social media on body image perception. Additionally, the findings showed that male amateur and professional football players were more likely than female football players to compare their physique to images seen online. In the control group of men, none of the respondents reported such comparisons. This suggests that men, both professional and amateur athletes, are more susceptible to the influence of social media on body image perception.

The portrayal of body images on social media is often misleading and unrealistic, which has a significant impact on individuals, especially athletes who are frequently exposed to such content. This can lead to distorted body image perceptions and increased pressure to conform to unattainable ideals. For athletes, whose performance and success are often associated with their physical appearance, this influence can be particularly harmful, potentially contributing to the development of EDs and body dissatisfaction. Therefore, it is crucial to address the impact of these false representations in discussions on athletes’ mental health and to develop strategies that promote a healthier and more realistic body image within the sports community.

The study also found a significant relationship between body dissatisfaction and the risk of EDs. Both professional and amateur athletes who were dissatisfied with their appearance and wanted to change many aspects of their bodies were more at risk for EDs. Among professional football players, 47.6% of those dissatisfied with their appearance showed an elevated risk of EDs, compared to 20.0% of those who were fully satisfied with their appearance. These findings suggest that body dissatisfaction, driven by both specific sports requirements and social comparisons, is a key factor in the development of EDs. This has also been confirmed in previous studies [28,29].

Interestingly, the analysis suggested that predictors such as body mass (current, lowest, highest, and ideal) had limited significance in explaining the risk of EDs assessed by the EAT-26, as indicated by the low R^2^ values (5.65% and 3.52% for models related to sex and sports level, respectively). This may indicate that biological factors, such as weight, are only a small component of the risk for EDs, whereas other, more difficult-to-measure factors, such as stress, social norms, peer pressure, or body image, may play a much more significant role. The lack of significant effects of variables related to body mass suggests that a broader consideration of psychological factors is necessary in assessing the risk of EDs, such as self-esteem, stress levels, or body perception, as well as environmental factors, such as social support and expectations from family or coaches. This points to the complexity of the issue of EDs and suggests that risk assessment should be more holistic, taking into account both internal and external factors that may affect well-being and eating-related behaviors.

The differences observed between professional and amateur athletes may be attributed to several key mechanisms that require deeper understanding. Professional athletes often face greater pressure to achieve and stricter requirements to maintain a specific physical form, which can lead to an increased risk of EDs. High levels of stress, more frequent comparisons with other athletes, and intensive use of social media, where unrealistic body standards are promoted, may further exacerbate these risks. As a result, professional athletes may be more vulnerable to the negative psychological effects associated with body image and the pressure for athletic success. The lack of significant differences between amateur and professional athletes in some areas may indicate that both are similarly at risk, potentially due to similar attitudes toward their bodies and the need to maintain adequate physical condition. Our findings highlight the need for more individualized psychological support programs and preventive interventions that are tailored to the specific challenges and pressures faced by athletes.

In light of the obtained results, it is crucial to conduct a more in-depth analysis of the differences between the studied groups, considering sport level, sex, and the intensity of social media use, to better understand the specificity of risk factors in the context of athletes. These differences may play a key role in shaping the prevalence of EDs and should be taken into account when designing future preventive interventions. Our findings suggested the need to implement targeted educational programs and preventive interventions aimed at both athletes and coaches, with the goal of raising awareness about the risk of EDs and promoting healthy eating habits and a positive body image.

Future research should focus on assessing the long-term effects of exposure to social media that promote unattainable aesthetic standards and its impact on body perception and EDs in different groups of athletes. Moreover, it is essential to evaluate the effectiveness of the proposed interventions to determine which strategies are most effective in reducing the risk of EDs among athletes at different levels of advancement. Implementing such measures will not only contribute to improving the mental and physical health of athletes but also provide a better understanding of the dynamics of risk factors in the modern sports environment, which could have practical significance for other athletic populations worldwide.

The findings of the study generally confirmed the initial hypotheses. A significant association was found between the risk of developing EDs and key variables, such as sport level, sex, and the frequency of social media use. As predicted, professional athletes, particularly women, were shown to be more at risk for eating disorders than amateur athletes. Specifically, professional female football players demonstrated an elevated risk of EDs, consistent with the hypothesis that female athletes are more susceptible to such disorders. Moreover, the hypothesis that social media use plays a key role in influencing body image and the risk of eating disorders was also confirmed. The study showed that individuals with greater exposure to social media, particularly athletes who compared their bodies with others online, exhibited a higher likelihood of ED risk. This supports the expected relationship between increased social media activity and greater vulnerability to EDs, in line with the hypothesis.

Furthermore, the control group (non-athletes) exhibited a lower prevalence of ED risk compared to both professional and amateur athletes. This finding reinforces the hypothesis that engagement in competitive sports, particularly at the professional level, presents a significant risk factor for the development of EDs, as these athletes are subjected to more intense body-related pressures than non-athletes.

Exercise and sport are widely recognized as essential components of a healthy lifestyle, promoting physical fitness, mental well-being, and overall quality of life. However, the study’s findings suggested that for many individuals, particularly those who are more susceptible to the influence of social media, body image concerns are a significant motivator for engaging in these activities. This focus on appearance rather than health can lead to problematic behaviors, such as excessive exercise, unhealthy dieting, and the use of performance-enhancing substances, all of which can have detrimental effects on both physical and mental health. It is crucial to address these concerns by promoting a holistic view of exercise and sport, emphasizing the intrinsic benefits of physical activity, such as improved health, strength, and well-being, rather than solely focusing on achieving an idealized body image. Educating individuals on the diverse benefits of exercise and challenging unrealistic body standards perpetuated by social media are essential steps toward fostering a healthier relationship with physical activity.

In conclusion, these results indicate that focusing solely on physical factors, such as body mass, is not sufficient in assessing and managing the risk of EDs. It is crucial to consider the social, psychological, and cultural determinants that may have a stronger impact on the risk of developing such disorders, particularly among women and athletes. By adopting a more holistic approach that encompasses both internal and external factors, interventions can be more effectively designed to address the complex interplay of influences contributing to EDs. This approach will not only enhance the well-being of athletes but also contribute to a healthier sports culture that values mental health and realistic body standards.

The study’s findings suggest an urgent need to develop innovative and targeted intervention strategies that go beyond the existing literature and are tailored to the specific findings of this study. Firstly, it is necessary to introduce educational programs that not only raise awareness about EDs but also actively engage athletes in shaping a healthy body image. This can be achieved through workshops and training focused on building psychological resilience, self-acceptance, and critical evaluation of media content.

Moreover, due to the significant influence of social media on body perception, we propose developing media campaigns that promote realistic and diverse body images. Collaborating with influencers and opinion leaders in the sports community can help spread positive messages and encourage healthier habits.

Another innovative approach is the integration of digital technologies into interventions. Mobile applications and online platforms can be used to monitor athletes’ well-being, provide personalized nutritional guidance, and offer real-time psychological support. Such tools can also enable quick identification of individuals at risk and direct them to specialized help. It is also important to involve coaches and support staff in the intervention process. Training coaches to recognize early signs of EDs and promote a healthy sports environment can significantly impact risk reduction. Implementing club and organizational policies that support mental health and counteract unhealthy eating practices is another step toward creating a safer environment for athletes.

Finally, we propose establishing collaborations between various entities—sports organizations, educational institutions, healthcare services, and media—to create an integrated approach to the issue of EDs in sports. Such partnerships can facilitate the exchange of knowledge, resources, and best practices, enhancing the effectiveness of interventions.

Future research should focus on assessing the effectiveness of these innovative intervention strategies in different groups of athletes. Particular emphasis should be placed on the long-term effects of these interventions and their impact on improving athletes’ mental and physical health. Additionally, it is essential to investigate the mechanisms through which social media influences body perception and the development of EDs to develop more precise and effective methods of counteracting these negative influences.

### Strengths and Limitations

It is worth emphasizing that this is the first study to comprehensively analyze the risk of ED among the population of Polish football players, including both professional and amateur athletes, as well as a control group of physically inactive individuals. The novelty of our study lies in the inclusion and detailed analysis of variables that had not previously been examined in the context of ED risk among Polish football players. For the first time in the literature, the relationship between ED risk and subjective comparisons of one’s body to photos posted on social media, the impact of elimination diets, and sources of nutritional knowledge has been presented. The study also considered variables related to the influence of social media, which constitutes a valuable contribution to research on external factors that may increase ED risk among athletes. The study also included additional, in-depth analyses, such as a linear regression analysis of ED risk about the lowest, highest, and subjectively perceived ideal body weight in adulthood, further reinforcing the innovative nature of our work. Our results provide new insights into the role of these factors in shaping ED risk, making this study a significant contribution to the literature on athletes’ mental and physical health.

One of the limitations of this study is its cross-sectional nature, which prevented establishing cause-and-effect relationships. Additionally, the reliance on self-reported survey data can introduce recall bias or intentional distortion by participants, particularly when dealing with sensitive issues, such as EDs and body image. The study also lacks objective laboratory or field measurements, which could have provided valuable data regarding the potential physiological consequences of eating disorders in athletes, such as their impact on muscle mass, fat percentage, and overall physical performance. Future studies should incorporate more precise physiological assessments, such as body composition analysis using techniques such as dual-energy X-ray absorptiometry (DXA) or bioelectrical impedance (BIA), to measure fat mass and lean/muscle mass, which would offer a more comprehensive understanding than BMI alone. It would be beneficial to investigate whether EDs impair aspects of football-specific performance, such as endurance, strength, agility, or decision-making, which are critical for success in the sport. The inclusion of performance metrics in future research could provide deeper insights into the functional consequences of disordered eating among athletes. Lastly, the study focused exclusively on football players and non-athletes in Poland, which limits the generalizability of the findings to other populations and cultures. The relatively small sample size may also not fully reflect the scope of the problem, indicating the need for larger, more diverse studies in the future.

## 5. Conclusions

The findings of this study underscored the critical importance of identifying risk factors associated with the development of EDs among both professional and amateur football players, as well as non-athletes. The results indicated that professional athletes, particularly women, exhibited a heightened susceptibility to EDs, reinforcing the significant roles of sex and sports level in influencing the risk of these disorders.

Additionally, the study highlighted the profound impact of social media on body perception, with athletes who frequently engage in comparisons of their physiques with others online showing an increased likelihood of experiencing ED risk.

These insights revealed the necessity for targeted interventions that address both the mental health of athletes, and the pressures imposed by social media. Sports organizations should establish psychological support programs aimed at mitigating the emphasis on idealized body standards and fostering a healthier body image. It is also essential to develop sex-specific interventions, as the study indicates that women and men face distinct pressures and risks associated with body image and EDs. For women, support should concentrate on alleviating pressures related to aesthetic ideals, while men could benefit from enhanced education on the dangers of body comparisons and unhealthy eating behaviors.

Moreover, the results suggested that professional athletes endure greater pressures compared to amateurs, likely due to heightened competitive demands and expectations regarding body weight and physical performance. Consequently, there is a pressing need to implement more comprehensive support systems for professional athletes, particularly those at an elevated risk for EDs.

## Figures and Tables

**Figure 1 nutrients-16-03470-f001:**
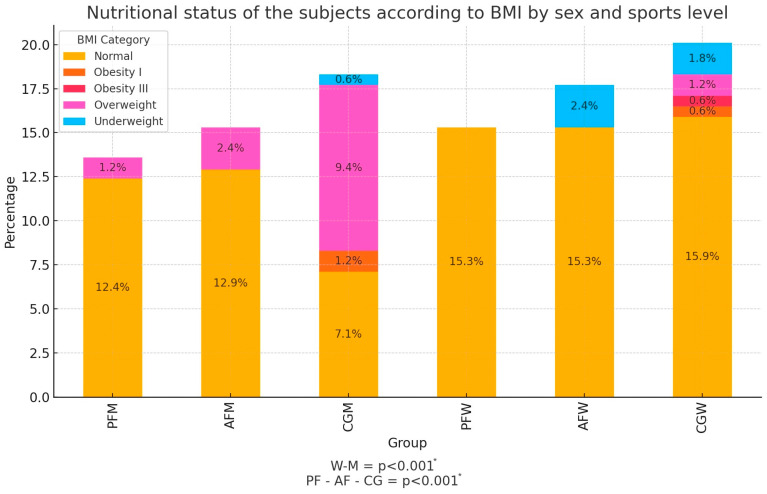
Nutritional status of the subjects according to the interpretation of BMI by sex and sports level (n = 170). M—male; W—female; PF—professional football level; AF—amateur football level; CG—control group; PFM—male professional football player; PFW—female professional football player; AFM—male amateur football player; AFW—female amateur football player; CGM—male control group; CGW—female control group; * = *p* < 0.05.

**Figure 2 nutrients-16-03470-f002:**
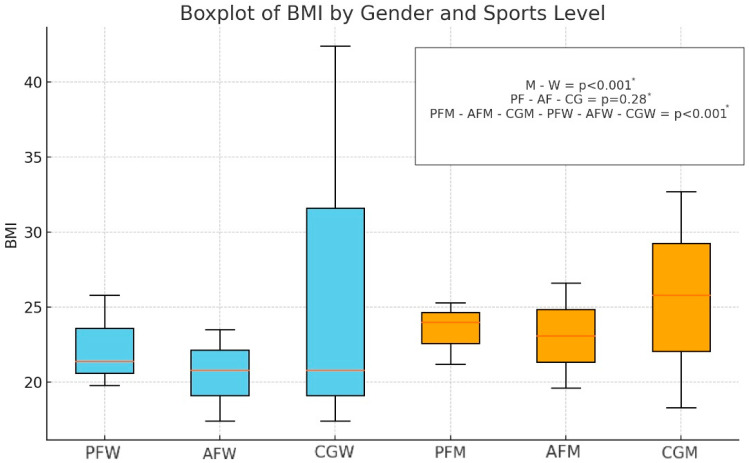
Distribution of BMI values by sex and sports level (n = 170). M—male; W—female; PF—professional football level; AF—amateur football level; CG—control group; PFM—male professional football player; PFW—female professional football player; AFM—male amateur football player; AFW—female amateur football player; CGM—male control group; CGW—female control group; * = *p* < 0.05.

**Table 1 nutrients-16-03470-t001:** Interpretation of BMI compared to norms for age [21].

Age	18	19	20	>20
BMI male	18.5	19.0	19.5	20.0
BMI female	18.0	18.0	18.5	19.0

**Table 2 nutrients-16-03470-t002:** Characteristics of the study group (n = 170).

	Age (Years) (X ± SD)	Height (cm) (X ± SD)	Body Mass (kg) (X ± SD)	BMI (kg/m^2^) (X ± SD)
Total (n = 170)	24.3 ± 4.20	172.0 ± 8.86	68.0 ± 13.2	22.7 ± 3.26
M (n = 80)	24.8 ± 4.20	179.0 ± 6.47	77.7 ± 9.65	24.2 ± 2.66
PFM (n = 23)	25.7 ± 4.98	179.0 ± 5.87	75.5 ± 7.82	23.6 ± 1.26
AFM (n = 26)	24.4 ± 2.58	180.0 ± 5.53	76.0 ± 8.97	23.4 ± 1.85
CGM (n = 31)	24.4 ± 4.71	179. 1 ± 7.69	80.8 ± 10.80	25.2 ± 3.57
*p*-value	0.513	0.802	0.091	0.050
W (n = 90)	23.9 ± 4.18	166.0 ± 5.44	59.4 ± 9.31	21.4 ± 3.22
PFW (n = 26)	23.0 ± 3.46	167.0 ± 6.63	60.4 ± 6.75	21.5 ± 1.29
AFW (n = 30)	24.3 ± 4.47	167.0 ± 4.52	58.1 ± 5.68	20.6 ± 1.87
CGW (n = 34)	24.1 ± 4.43	164.0 ± 4.82	59.9 ± 13.0	22.1 ± 4.74
*p*-value	0.378	0.040 *	0.374	0.056

M—male; W—female; PFM—male professional football player; PFW—female professional football player; AFM—male amateur football player; AFW—female amateur football player; CGM—male control group; CGW—female control group; * = *p* < 0.05.

**Table 3 nutrients-16-03470-t003:** Sources of nutritional knowledge by sex and sports level (n = 170).

Group	Dietitian, n (%)	Internet, n (%)	Scientific Literature, n (%)	I Do Not Obtain (Knowledge), n (%)	From Other Players, n (%)	From Friends, n (%)	Family, n (%)	Coach, n (%)	*p*-Value
Total (n = 170)	38 (22.4)	84 (49.4)	4 (2.4)	20 (11.8)	5 (2.9)	11 (6.5)	1 (0.6)	7 (4.1)	
M (n = 80)	19 (23.8)	34 (42.5)	1 (1.3)	13 (16.3)	3 (3.8)	5 (6.3)	0	5 (6.3)	<0.01 *
PFM (n = 23)	16 (69.6)	4 (17.4)	0 (0.0)	0 (0.0)	0 (0.0)	1 (4.3)	0 (0.0)	2 (8.7)	
AFM (n = 26)	1 (3.8)	16 (61.5)	1 (3.8)	0 (0.0)	3 (11.5)	2 (7.7)	0 (0.0)	3 (11.5)	
CGM (n = 31)	2 (6.5)	14 (45.2)	0 (0.0)	13 (41.9)	0 (0.0)	2 (6.5)	0 (0.0)	0 (0.0)	
W (n = 90)	19 (21.1)	50 (55.6)	3 (3.3)	7 (7.8)	2 (2.2)	6 (6.7)	1 (1.1)	2(2.2)	<0.01 *
PFW (n = 26)	4 (15.4)	12 (46.2)	1 (3.8)	0 (0.0)	2 (7.7)	4 (15.4)	1 (3.8)	2 (7.7)	
AFW (n = 30)	13 (43.3)	17 (56.7)	0 (0.0)	0 (0.0)	0 (0.0)	0 (0.0)	0 (0.0)	0 (0.0)	
CGW (n = 34)	2 (5.9)	21 (61.8)	2 (5.9)	7 (20.6)	0 (0.0)	2 (5.9)	0 (0.0)	0 (0.0)	
	M vs. W (*p* = 0.345) PF vs. AF vs. CG (*p* < 0.01 *)								

M—male; W—female; PF—professional football level; AF—amateur football level; CG—control group; PFM—male professional football player; PFW—female professional football player; AFM—male amateur football player; AFW—female amateur football player; CGM—male control group; CGW—female control group; * = *p* < 0.05.

**Table 4 nutrients-16-03470-t004:** Information regarding social media activity, body shape comparisons with other images on social media, and body satisfaction among the study group (n = 170).

Group	Total (n = 170)	M (n = 80)	PFM (n = 23)	AFM (n = 26)	CGM (n = 31)	*p*-Value	W (n = 90)	PFW (n = 26)	AFW (n = 30)	CGW (n = 34)	*p*-Value
Time of use of social media during the day, n (%)											
Up to 1 h, n (%)	5 (2.9)	3 (3.8)	1 (4.3)	2 (7.7)	0	0.046 *	2 (2.2)	2 (7.7)	0	0	0.136
1–2 h, n (%)	44 (25.9)	18 (22.5)	5 (21.7)	9 (34.6)	4 (12.9)		26 (28.9)	10 (38.5)	8 (26.7)	8 (23.5)	
2–3 h, n (%)	70 (41.2)	32 (40.0)	5 (21.7)	10 (38.5)	17 (54.8)		38 (42.2)	6 (23.1)	14 (46.7)	18 (52.9)	
Above 3 h, n (%)	51 (30.0)	27 (33.8)	12 (52.2)	5 (19.2)	10 (32.3)		24 (26.7)	8 (30.8)	8 (26.7)	8 (23.5)	
M vs. W (*p* = 0.623) PF vs. AF vs. CG (*p* = 0.016 *)											
The most common type of application, n (%)											
Facebook, n (%)	21 (12.4)	9 (11.3)	0	7 (26.9)	2 (6.5)	<0.001 *	12 (13.3)	1 (3.8)	8 (26.7)	3 (8.8)	<0.001 *
Instagram, n (%)	90 (59.2)	44 (55.0)	20 (87.0)	9 (34.6)	15 (48.4)		46 (51.1)	22 (84.6)	5 (16.7)	19 (55.9)	
TikTok, n (%)	33 (19.4)	14 (17.5)	3 (13.0)	5 (19.2)	6 (19.4)		19 (21.1)	2 (7.7)	5 (16.7)	12 (35.3)	
X (Twitter), n (%)	10 (5.9)	5 (5.6)	0	5 (19.2)	0		5 (5.6)	1 (3.8)	4 (13.3)	0	
YouTube, n (%)	16 (9.4)	8 (10.0)	0	0	8 (25.8)		8 (8.9)	0	8 (26.7)	0	
M vs. W (*p* = 0.958) PF vs. AF vs. CG (*p* < 0.001 *)											
Comparing body image to social media photos, n (%)											
No, never, n (%)	78 (46.5)	47 (58.8)	6 (26.1)	19 (73.1)	22 (71.0)	<0.01 *	32 (35.6)	12 (46.2)	14 (46.7)	6 (17.6)	<0.01 *
Yes, sometimes, n (%)	54 (31.8)	20 (25.0)	9 (39.1)	2 (7.7)	9 (29.0)		34 (37.8)	12 (46.2)	12 (40.0)	10 (29.4)	
Yes, often, n (%)	37 (21.8)	13 (16.3)	8 (34.8)	5 (19.2)	0		24 (26.7)	2 (7.7)	4 (13.3)	18 (52.9)	
M vs. W (*p* = 0.010 *) PF vs. AF vs. CG (*p* < 0.100)											
Satisfied with the appearance of body, n (%)											
I am not satisfied, there are many things I would like to change, n (%)	41 (24.1)	16 (20.0)	4 (17.4	5 (19.2)	7 (22.6)	0.110	25 (27.8)	4 (15.4)	9 (30.0)	12 (35.3)	0.081
Yes, but I would change a few things about my appearance, n (%)	91 (53.5)	42 (52.5)	8 (34.8)	17 (65.4)	17 (54.8)		49 (54.4)	13 (50.0)	18 (60.0)	1 (52.9)	
Yes, I wouldn’t change a thing, n (%)	38 (22.4)	22 (27.5)	11 (47.8)	4 (15.4)	7 (22.6)		16 (17.8)	9 (34.6)	3 (10.0)	4 (11.8)	
M vs. W (*p* = 0.237) PF vs. AF vs. CG (*p* = 0.006 *)											

M—male; W—female; PF—professional football level; AF—amateur football level; CG—control group; PFM—male professional football player; PFW—female professional football player; AFM—male amateur football player; AFW—female amateur football player; CGM—male control group; CGW—female control group; * = *p* < 0.05.

**Table 5 nutrients-16-03470-t005:** Information on current, lowest, highest, and subjectively considered ideal body mass in the study group (n = 170).

Group	Body Mass; Current	Body Mass; Lowest	Body Mass; Highest	Body Mass; Ideal	
Total (n = 170)	68.0 ± 13.2	61.8 ± 13.1	71.8 ± 15.6	65.7 ± 13.0	
M (n = 80)	77.7 ± 9.65	71.0 ± 12.1	80.9 ± 14.9	75.9 ± 11.4	Current–Highest, *p* < 0.001 * Current–Lowest, *p* < 0.001 * Current–Ideal, *p* = 0.322 (χ2 = 178; df = 3; *p* < 0.001)
PFM (n = 23)	75.5 ± 7.82	68.6 ± 16.51	74.2 ± 17.62	71.9 ± 16.78	
AFM (n = 26)	76.0 ± 8.97	70.4 ± 8.14	81.9 ± 12.92	75.0 ± 8.18	
CGM (n = 31)	80.8 ± 10.80	73.2 ± 11.12	85.2 ± 12.66	79.1 ± 7.46	
*p*-value	0.091	0.049 *	0.419	0.092	
W (n = 90)	59.4 ± 9.31	53.7 ± 7.26	63.6.7 ± 10.96	56.7 ± 5.28	Current–Highest, *p* < 0.001 * Current–Lowest, *p* < 0.001 * Current–Ideal, *p* < 0.001 * (χ2 = 223; df = 3; *p* < 0.001)
PFW (n = 26)	60.4 ± 6.75	56.5 ± 6.73	63.5 ± 8.36	59.0 ± 6.10	
AFW (n = 30)	58.1 ± 5.68	53.3 ± 4.24	62.0 ± 5.02	55.6 ± 3.40	
CGW (n = 34)	59.9 ± 13.0	51.9 ± 9.07	65.0 ± 15.63	55.9 ± 5.54	
*p*-value	0.374	*p* = 0.056	*p* = 0.450	*p* = 0.049 *	
	M vs. W (*p* < 0.01 *) PF vs. AF vs. CG (*p* = 0.390)	M vs. W (*p* < 0.01 *) PF vs. AF vs. CG (*p* = 0.899)	M vs. W (*p* < 0.01 *) PF vs. AF vs. CG (*p* = 0.128)	M vs. W (*p* < 0.01 *) PF vs. AF vs. CG (*p* = 0.649)	

M—male; W—female; PF—professional football level; AF—amateur football level; CG—control group; PFM—male professional football player; PFW—female professional football player; AFM—male amateur football player; AFW—female amateur football player; CGM—male control group; CGW—female control group; * = *p* < 0.05; χ2—Friedman test; df—degrees of freedom.

**Table 6 nutrients-16-03470-t006:** Summary of ED risk estimation (EAT-26; n = 170).

	Part A		Part B		Part C		Entire	
EAT-26	No Risk	Elevated Risk	No Risk	Elevated Risk	No Risk	Elevated Risk	No Risk	Elevated Risk
Total (n = 170)	153 (90.0)	17 (10.0)	142 (83.5)	28 (16.5)	154 (90.6)	16 (9.4)	113 (66.5)	57 (33.5)
M (n = 80)	77 (96.3)	3 (3.8)	66 (82.5)	14 (17.5)	79 (98.8)	1 (1.3)	62 (77.5)	18 (22.5)
PFM (n = 23)	23 (100)	0	13 (56.5)	10 (43.5)	23 (100)	0	13 (56.5)	10 (43.5)
AFM (n = 26)	23 (88.5)	3 (11.5)	24 (92.3)	2 (7.7)	25 (96.2)	1 (3.8)	21 (80.8)	5 (19.2)
CGM (n = 31)	31 (100)	0	29 (93.5)	2 (6.5)	31 (100)	0	28 (90.3)	3 (9.7)
W (n = 90)	76 (84.4)	14 (15.6)	76 (84.4)	14 (15.6)	75 (83.3)	14 (16.7)	51 (56.7)	39 (43.3)
PFW (n = 26)	23 (88.5)	3 (11.5)	18 (69.2)	8 (30.)	24 (92.3)	2 (7.7)	15 (57.7)	11 (42.3)
AFW (n = 30)	21 (70.0)	9 (30.0)	30 (100)	0	22 (73.3)	8 (26.7)	14 (46.7)	16 (53.3)
CGW (n = 34)	32 (94.1)	2 (5.9)	38 (82.4)	6 (17.6)	29 (85.3)	5 (14.7)	22 (64.7)	12 (35.3)
	M vs. W (*p* = 0.010 *) PF vs. AF vs. CG (*p* = 0.002 *)		M vs. W (*p* = 0.733) PF vs. AF vs. CG (*p* < 0.001 *)		M vs. W (*p* < 0.001 *) PF vs. AF vs. CG (*p* < 0.092)		M vs. W (*p* < 0.004 *) PF vs. AF vs. CG (*p* < 0.064)	

M—male; W—female; PF—professional football level; AF—amateur football level; CG—control group; PFM—male professional football player; PFW—female professional football player; AFM—male amateur football player; AFW—female amateur football player; CGM—male control group; CGW—female control group; * = *p* < 0.05.

**Table 7 nutrients-16-03470-t007:** Assessment of ED risk evaluated based on the interpretation of the EAT-26 scale, considering social media activity, comparison of body shape with other images on social media, and body satisfaction in the study group (n = 170).

Group	Total (n = 170)		M (n = 80)		W (n = 90)		PF (n = 49)		AF (n = 56)		CG (n = 65)	
	No Risk	Elevated Risk	No Risk	Elevated Risk	No Risk	Elevated Risk	No Risk	Elevated Risk	No Risk	Elevated Risk	No Risk	Elevated Risk
	Time of use of social media during the day, n (%)											
Up to 1 h, n (%)	3 (60.0)	3 (60.0)	2 (66.7)	1 (33.3)	0	2 (100)	0	3 (100)	2 (100)	0	0	0
1–2 h, n (%)	13 (29.5)	13 (29.5)	14 (77.8)	4 (22.2)	17 (65.4)	9 (34.6)	9 (60.0)	6 (40.0)	13 (76.5)	4 (23.4)	9 (75.0)	3 (25.0)
2–3 h, n (%)	19 (27.1)	19 (27.1)	27 (84.4)	5 (15.6)	24 (63.2)	14 (36.8)	8 (72.7)	3 (27.2)	14 (58.3)	10 (41.7)	29 (75.0)	6 (17.1)
Above 3 h, n (%)	22 (43.1)	22 (43.1)	19 (70.4)	8 (29.6)	10 (41.7)	14 (58.3)	11 (55.0)	9 (45.0)	6 (46.2)	7 (53.8)	12 (66.7)	6 (33.3)
*p*-value	0.152		0.603		0.099		0.159		0.233		-	
Cramer’s V	0.176		0.152		0.264		0.325		0.276		-	
	The most common type of application, n (%)											
Facebook, n (%)	17 (81.0)	4 (19.0)	9 (100.0)	0 (0.0)	8 (66.7)	4 (33.3)	1 (100)	0 (0.0%)	12 (80.0)	3 (20.0)	4 (80.0)	1 (20.0)
Instagram, n (%)	60 (66.7)	30 (33.3)	31 (70.5)	13 (29.5)	29 (63.0)	17 (37.0)	24 (57.1)	18 (42.9)	7 (50.0)	7 (50.0)	29 (85.3)	5 (14.7)
TikTok, n (%)	24 (72.7)	9 (27.3)	13 (92.9)	1 (7.1)	11 (57.9)	8 (42.1)	3 (60.0)	2 (40.0)	10 (100.0)	0 (0.0)	11 (61.1)	7 (38.9)
X (Twitter), n (%)	3 (30.0)	7 (70.0)	3 (60.0)	2 (40.0)	0 (0.0)	5 (100.0)	0 (0.0)	1 (100)	3 (33.3)	6 (66.7)	0	0
YouTube, n (%)	9 (56.3)	7 (43.8)	6 (75.0)	2 (25.0)	3 (37.5)	5 (62.5)	0	0	3 (37.5)	5 (62.5)	6 (75.0)	2 (25.0)
*p*-value	0.055		0.061		0.155		-		0.006 *		-	
Cramer’s V	0.234		0.289		0.316		-		0.505		-	
	Comparing body image to social media photos, n (%)											
No, never, n (%)	62 (78.5)	17 (21.5)	41 (87.2)	6 (12.8)	21 (65.6)	11 (34.4)	12 (66.7)	6 (33.3)	25 (75.8)	8 (24.2)	25 (89.3)	3 (10.7)
Yes, sometimes, n (%)	17 (45.9)	20 (54.1)	4 (30.8)	9 (69.2)	13 (54.2)	11 (45.8)	3 (30.0)	7 (70.0)	3 (33.3)	6 (66.7)	11 (61.1)	7 (38.9)
Yes, often, n (%)	34 (63.0)	20 (37.0)	17 (85.0)	3 (15.0)	17 (50.0)	17 (50.0)	13 (61.9)	8 (38.1)	7 (50.0)	7 (50.0)	14 (73.7)	5 (26.3)
*p*-value	0.002 *		<0.001 *		0.423		0.144		0.036 *		0.80	
Cramer’s V	0.270		0.493		0.138		0.281		0.345		0.279	
	Satisfied with the appearance of body, n (%)											
I am not satisfied, there are many things I would like to change, n (%)	62 (68.1)	29 (31.9)	32 (76.2)	10 (23.8)	30 (61.2)	19 (38.8)	11 (52.4)	10 (47.6)	23 (65.7)	12 (34.3)	28 (80.0)	7 (20.0)
Yes, but I would change a few things about my appearance, n (%)	21 (51.2)	20 (48.8)	12 (75.0)	4 (25.0)	9 (36.0)	16 (64.0)	4 (50.0)	4 (50.0)	5 (35.7)	9 (64.3)	12 (63.2)	7 (36.8)
Yes, I wouldn’t change a thing, n (%)	30 (78.9)	8 (21.1)	18 (81.8)	4 (18.2)	12 (75.0)	4 (25.0)	13 (65.0)	7 (35.0)	7 (100.0)	0 (0.0)	10 (90.9)	1 (9.1)
*p*-value	0.03 *		0.846		0.031 *		0.649		0.013 *		0.180	
Cramer’s V	0.204		0.065		0.278		0.133		0.393		0.230	

M—man; W—women; PF—professional football level; AF—amateur football level; CG—control group; PFM—male professional football player; PFW—female professional football player; AFM—male amateur football player; AFW—female amateur football player; CGM—male control group; CGW—female control group; * = *p* < 0.05.

**Table 8 nutrients-16-03470-t008:** Linear regression analysis of body mass predictors and EAT-26 score in sex groups (n = 170).

Model Coefficients—EAT-26 Score; R = 0.238; R2 = 0.0565				
Predictor	Estimate	SE	t	*p*
Body mass; Current	0.1059	0.0907	1.167	0.245
Body mass; Lowest	−0.0772	0.1030	−0.749	0.455
Body mass; Highest	−0.1184	0.1348	−0.879	0.381
Body mass; Ideal	0.2256	0.1352	1.669	0.097
Gender (M–W)	−5.3986	1.9020	−2.838	0.005 *

M—man; W—women; SD—standard deviations, t—t-statistics; * = *p* < 0.05.

**Table 9 nutrients-16-03470-t009:** Linear regression analysis of body mass predictors and EAT-26 score in sports level groups (n = 170).

Model Coefficients—EAT-26 Score; R = 0.178; R2 = 0.0352				
Predictor	Estimate	SE	t	*p*
Body mass; Current	−0.0409	0.0770	−0.531	0.596
Body mass; Lowest	0.1331	0.0991	1.342	0.181
Body mass; Highest	−0.1921	0.1460	−1.315	0.190
Body mass; Ideal	0.0419	0.1140	0.367	0.714
Gender (AF–PF)	−0.9080	1.3704	−0.663	0.509
Gender (CG–PF)	−2.7545	1.4058	−1.959	0.049 *

PF—professional football level; AF—amateur football level; CG—control group; SD—standard deviations, t—t-statistics; * = *p* < 0.05.

## Data Availability

The raw data supporting the conclusions of this article will be made available by the authors upon request.

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
