# Peer review of "Assessment of Eating Disorder Risk According to Sport Level, Sex, and Social Media Use among Polish Football Players: A Cross-Sectional Study"

_nutrients, 2024, doi:10.3390/nu16203470_

Round 1
Reviewer 1 Report (Previous Reviewer 1)
Comments and Suggestions for Authors
the paper needs novelty and depth to make a valid contribution to the existing literature. for this reason it cannot be accepted.
1. Originality of the results (or lack of novelty):
- The paper makes no new contributions to the existing literature. Many previous studies have already shown that female athletes in particular are more vulnerable to eating disorders due to aesthetic pressures and performance expectations. Furthermore, the negative influence of social media on body perception is well documented. It is necessary for the study to propose innovative results or a more in-depth analysis to justify publication.
2. Depth of analysis:
- Although the findings confirm known trends, there is a lack of in-depth discussion on the psychological and social mechanisms that differentiate the risk of eating disorders between professional, amateur and non-athlete athletes. A more detailed analysis of the specific factors contributing to these differences could offer new insights.
3. Practical impact of the conclusions
- The conclusions suggest the need for targeted interventions, but no new approaches or strategies are proposed over and above what is already in the literature. It would be helpful if the paper provided innovative guidelines or intervention strategies based on the specific findings of the study.
4. Use of data and methodology:
- The methodology is clear and well structured, but the results obtained do not seem to exploit the full potential of the data collected. A re-analysis of the data with more advanced approaches or the addition of less explored variables, e.g. the impact of different social platforms in more detail or the influence of specific cultural factors, could be useful.
5. Need for more contextualisation:
- The paper could benefit from better contextualisation than the existing literature. Although the topic of eating disorders in athletes is well known, a more nuanced discussion that puts the findings of the study in the context of more recent research could make it more relevant.
Author Response
Dear Reviewer,
Thank you very much for re-evaluating our study, I hope you find the answers satisfactory.
- Originality of the results (or lack of novelty):
- The paper makes no new contributions to the existing literature. Many previous studies have already shown that female athletes in particular are more vulnerable to eating disorders due to aesthetic pressures and performance expectations. Furthermore, the negative influence of social media on body perception is well documented. It is necessary for the study to propose innovative results or a more in-depth analysis to justify publication.
Thank you for your detailed comments on my article. I would like to present several arguments in support of the innovation and value of the study.
First and foremost, I would like to emphasize that this study is one of the first to simultaneously assess the risk of eating disorders based on sport level (professional vs. amateur), the degree of social media use, and the inclusion of gender among the specific group of football players. The occurrence of eating disorders is dependent on the sport discipline, and while football is not considered a "high-risk" sport according to current literature, our research highlights an often-overlooked problem within this group.
Regarding cultural differences, more and more research indicates that nationality and cultural factors may influence the risk of eating disorders. An example is our previous research, which analyzes the impact of sociocultural attitudes toward body image on football players from Poland and Turkey (Staśkiewicz-Bartecka, W., Aktaş, S., Zydek, G., Kardas, M., Kałuża, M., & Michalczyk, M. M. (2024). Eating disorder risk assessment and sociocultural attitudes toward body image among Polish and Turkish professional female football players. Frontiers in Nutrition, 11, 1456782) and the risk of orthorexia among football players from different countries (Staśkiewicz-Bartecka, W., Kalpana, K., Aktaş, S., Khanna, G. L., Zydek, G., Kardas, M., & Michalczyk, M. M. (2024). The Impact of Social Media and Socio-Cultural Attitudes toward Body Image on the Risk of Orthorexia among Female Football Players of Different Nationalities. Nutrients, 16(18), 3199). The submitted study is the first to thoroughly analyze the problem of eating disorders in the population of Polish football players. It was planned and conducted to precisely summarize the scale of the problem among Polish athletes compared to the general population.
In light of the above, we believe that the study makes a significant contribution to the existing literature, offering a comprehensive analysis of eating disorder risks in the population of Polish athletes, from the perspective of sport level, gender, and social media usage intensity.
However, we conducted an additional analysis, which we described further in our response to the review, in order to further enhance the innovation of the study.
- Depth of analysis:
- Although the findings confirm known trends, there is a lack of in-depth discussion on the psychological and social mechanisms that differentiate the risk of eating disorders between professional, amateur and non-athlete athletes. A more detailed analysis of the specific factors contributing to these differences could offer new insights.
The article analyzes specific psychological and social mechanisms that differentiate the risk of eating disorders between professional athletes, amateurs, and non-athletes. We highlight specific factors such as comparing one’s body with photos posted online and body image satisfaction, which is particularly important in today’s world where social media plays an increasingly significant role in shaping body perception, especially among young athletes.
Following your suggestion, we decided to conduct a more in-depth analysis of our data. In the study, we included the correlation between the lowest, highest, and subjectively perceived ideal body weight and the risk of eating disorders. To this end, we performed a linear regression analysis of the effect of the lowest, highest, and subjectively perceived ideal body weight on the EAT-26 test score.
- Practical impact of the conclusions
- The conclusions suggest the need for targeted interventions, but no new approaches or strategies are proposed over and above what is already in the literature. It would be helpful if the paper provided innovative guidelines or intervention strategies based on the specific findings of the study.
Thank you very much for your suggestion, in line with your comment we have completed the missing information in the discussion section of the survey.
“The study's findings suggest an urgent need to develop innovative and targeted intervention strategies that go beyond the existing literature and are tailored to the specific findings of this study. Firstly, it is necessary to introduce educational programs that not only raise awareness about EDs but also actively engage athletes in shaping a healthy body image. This can be achieved through workshops and training focused on building psychological resilience, self-acceptance, and critical evaluation of media content.
Moreover, due to the significant influence of social media on body perception, we propose developing media campaigns that promote realistic and diverse body images. Collaborating with influencers and opinion leaders in the sports community can help spread positive messages and encourage healthier habits.
Another innovative approach is the integration of digital technologies into interventions. Mobile applications and online platforms can be used to monitor athletes' well-being, provide personalized nutritional guidance, and offer real-time psychological support. Such tools can also enable quick identification of individuals at risk and direct them to specialized help.
It is also important to involve coaches and support staff in the intervention process. Training coaches to recognize early signs of EDs and promote a healthy sports environment can significantly impact risk reduction. Implementing club and organizational policies that support mental health and counteract unhealthy eating practices is another step toward creating a safer environment for athletes.
Finally, we propose establishing collaborations between various entities—sports organizations, educational institutions, healthcare services, and media—to create an integrated approach to the issue of EDs in sports. Such partnerships can facilitate the exchange of knowledge, resources, and best practices, enhancing the effectiveness of interventions.”
- Use of data and methodology:
- The methodology is clear and well structured, but the results obtained do not seem to exploit the full potential of the data collected. A re-analysis of the data with more advanced approaches or the addition of less explored variables, e.g. the impact of different social platforms in more detail or the influence of specific cultural factors, could be useful.
Thank you very much for your valuable suggestion. We have added an additional linear regression analysis as well as an analysis of individual social media and their impact on EDs risk.
- Need for more contextualisation:
- The paper could benefit from better contextualisation than the existing literature. Although the topic of eating disorders in athletes is well known, a more nuanced discussion that puts the findings of the study in the context of more recent research could make it more relevant.
Corrected as suggested.
Reviewer 2 Report (Previous Reviewer 2)
Comments and Suggestions for Authors
Authors answered all my comments. Therefore, the manuscript can be accepted in the present form.
Author Response
Thank you for accepting our study.
Reviewer 3 Report (Previous Reviewer 4)
Comments and Suggestions for Authors
Dear Authors,
Thank you for submitting the revised version of the manuscript. Great work!
I have only one suggestion for improvement: please consider adding asterisks (e.g., *) to indicate statistical differences in Figures 1 and 2. The figures should be self-explanatory and understandable to the reader without referring to the text. Additionally, would it be more appropriate to present observed counts as percentages in Figure 1?
Author Response
Dear Reviewer,
Thank you very much for your suggestions, we have corrected the graphs by marking the significance and indicated the percentages in Figure 1.
Round 2
Reviewer 1 Report (Previous Reviewer 1)
Comments and Suggestions for Authors
The study ‘Assessment of Eating Disorder Risk According to Sport Level, Sex, and Social Media Use Among Polish Football Players’ addresses a relevant topic, namely the assessment of eating disorder risk among athletes, with a particular focus on differences related to sport level, sex, and social media use. However, there are some methodological criticalities and innovativeness considerations that deserve to be discussed in order to strengthen the study's conclusions.
One methodological aspect that raises doubts is the use of different BMI thresholds for men and women. While there are physiological differences between the two sexes, the use of a differentiated BMI has no solid basis in the most rigorous scientific literature. BMI is already a standardised and much debated measure, as it does not distinguish between muscle mass and fat mass and does not take body composition into account. Adopting different thresholds for men and women could add further confusion and compromise the comparability of results. It would have been more appropriate to use other anthropometric measures or more advanced methodologies, such as body composition analysis (e.g. by means of bioimpedance analysis or DEXA), which would have provided a more precise and scientifically justified assessment of the participants' nutritional status.
Furthermore, the use of BMI as the sole indicator of eating disorder risk does not seem entirely appropriate, considering that BMI may underestimate or overestimate risk in certain populations, such as athletes, who often have greater muscle mass. The inclusion of more accurate measurements would have provided a more complete and reliable picture.
Despite the authors' revision in response to previous comments, significant doubts remain about the innovativeness of the study. Although the topic is topical and of interest, the work seems to replicate findings already widely known in the literature concerning eating disorders in athletes, especially with regard to the vulnerability of female athletes and the influence of social media. It lacks a truly original approach that could offer new perspectives or practical applications in the field of sports nutrition or eating disorder prevention.
Recommendations:
- I suggest revising the methodological approach related to BMI, abandoning gender-differentiated thresholds in favour of a more precise and representative measure of participants' health status.
- I also suggest better justifying the use of BMI as a tool to assess the risk of eating disorders in an athletic population, or considering alternative measures that are more relevant to this type of subject.
- It would be useful for further study to show how this study can make an innovative contribution to the existing literature.
In conclusion, the study presents interesting results, but the use of gender-specific BMI criteria is a methodological limitation. Furthermore, despite the revision, the study continues to show limited originality, which could reduce its impact in the scientific landscape.
Author Response
Dear Reviewer,
Thank you very much for your review of our study. I have carefully reviewed the suggestions you provided. However, I would like to clarify certain issues that indicate a misunderstanding of our study. You pointed out that the main methodological flaw of the study is the use of different BMI thresholds for men and women. I would like to emphasize that the nutritional status was assessed using WHO-recommended standards, which are the same for both men and women. Please refer to the study's methodology description and subpoint 2.3.1, where we state that we followed the WHO recommendations, reference number 20 from the cited literature. I have the impression that you were misled by subpoint 2.3.2, which describes the EAT-26 questionnaire. The EAT-26 questionnaire includes a BMI assessment based on age and gender, which is an integral part of this validated tool. Therefore, the suggestion cannot be implemented in the study and may indicate a misunderstanding of the research methodology. Below, I have included a link to the questionnaire and its standardization: https://www.eat-26.com/scoring/
Furthermore, we noted in the study's limitations that the BMI index has its limitations, and we explained this in detail in section 4.1. At the same time, I would like to highlight that we conducted a study that includes body composition obtained through BIA and its impact on the risk of eating disorders (ED). This study, describing our results, will be published in the Journal of Human Kinetics in January 2025. Comparing the body composition of professional football players, amateurs, and physically inactive individuals will show obvious differences in body composition. I would like to emphasize that the purpose of our study is to assess differences in ED risk depending on the level of athletic performance to obtain specific information on what differences exist between the general population and athletes practicing this discipline, also taking into account gender differences and the use of social media. The innovativeness of our research, following the editor's suggestion, has also been clearly described in section 4.1, where we emphasized its value. In response to your comments, we conducted additional in-depth analyses to further strengthen the novelty of our study. In particular, we conducted linear regression analysis concerning the risk of eating disorders in relation to the lowest, highest adult body weight, and subjectively perceived ideal body weight. No previous studies conducted on the population of football players have analyzed such variables in relation to ED risk. Additionally, we would like to highlight that our study includes variables that have not been previously analyzed, such as comparing one's body to images posted on social media and identifying the impact of elimination diets or nutritional knowledge sources on ED risk.
This study is one of the few that provides an in-depth analysis of the risk of eating disorders based on athletic performance level, gender, and the use of social media among Polish female football players. When publishing our previous studies (Staśkiewicz-Bartecka, W., Kalpana, K., Aktaş, S., Khanna, G. L., Zydek, G., Kardas, M., & Michalczyk, M. M. (2024). The Impact of Social Media and Socio-Cultural Attitudes toward Body Image on the Risk of Orthorexia among Female Football Players of Different Nationalities. Nutrients, 16(18), 3199.; Staśkiewicz-Bartecka, W., Aktaş, S., Zydek, G., Kardas, M., Kałuża, M., & Michalczyk, M. M. (2024). Eating disorder risk assessment and sociocultural attitudes toward body image among Polish and Turkish professional female football players. Frontiers in Nutrition, 11, 1456782.), reviewers suggested that future results should be compared with a population of physically inactive individuals; therefore, we decided to conduct a thorough analysis to fill this gap in the field of eating disorders in the Polish football population.
Once again, thank you for your constructive feedback. We hope that the additional analyses are satisfactory and emphasize the innovative aspects of our study.
This manuscript is a resubmission of an earlier submission. The following is a list of the peer review reports and author responses from that submission.
Round 1
Reviewer 1 Report
Comments and Suggestions for Authors
Thank you for the opportunity to review the paper.
There are several shortcomings in this manuscript.
1.Lack of originality: The study addresses topics already widely explored in the literature, such as eating disorders among athletes and the influence of social media on body image. Although the analysis of professional and amateur footballers may seem interesting, no truly innovative aspects or new methodologies emerge that add a significant contribution to the understanding of the phenomenon. Previous studies have already identified gender, sporting level and social media use as major risk factors for eating disorders, especially among women.
2.Methodological weakness: The methodological approach, although adequate for a descriptive data collection, does not provide sufficient analysis to support the claims made. The use of self-reported instruments, such as the EAT-26 test, introduces potential bias, and the lack of more objective or innovative measures limits the validity of the results. Furthermore, the sample, although representing athletes of both sexes and sporting levels, is limited geographically and culturally, which reduces the generalisability of the results.
3.Insufficient interpretation: The study's conclusions merely restate what is already known, without providing significant insights into possible practical implications or new research directions. Possible targeted interventions or preventive approaches to reduce the risk of eating disorders are not sufficiently explored, nor is a critical analysis of the specific factors that could influence the results in the various study groups proposed.
4. Inconsistencies in the results: Some of the differences observed between the professional and amateur groups are not adequately explored. There is a lack of clear discussion of the mechanisms leading to a higher risk in professionals, and the relationships between the data are not always supported by a robust statistical analysis. This weakens the reliability of the conclusions drawn by the authors.
Plagiarism. Pay attention to the plagiarism in the introduction and methods (see attachment)
Although the topic discussed is of relevance to public health and athletes, the study lacks sufficient originality, depth and methodological rigour to be published in its current form.

Reviewer 2 Report
Comments and Suggestions for Authors
Very interesting study. The authors study a topic of growing interest in nutrition and health. Especially, the impact of social networks on diet and body image perception. However, I have the following comments.
I. Comments:
1. Improve the wording of the study objective.
2. Improve the quality of the resolution of the figures.
3. The authors did not evaluate the subjects' diet, but it would be interesting to discuss this point. For example, what foods do they avoid eating?
4. The body images available on social networks may be false. This could be discussed by the authors.
5. Exercise and sport should be practiced as part of a healthy life. However, these (worrying) results would be demonstrating that body image would be a central aspect that motivates the practice of exercise and sport. Discuss.
6. What projections would this study have?
Reviewer 3 Report
Comments and Suggestions for Authors
The title is long and confusing. I suggest replacing “about” with “according to”. The term “gender” refers to a behavioral characteristic. I think you might have assessed “sex” in your study (which is a biological characteristic). I also suggest shortening the title if possible.
Line 139: I suggest adding a reference to the validation of your eating disorders questionnaire (the EAT-26).
Table 2: I suggest shortening this table by presenting the norms only for the age groups for the participants in your study.
In your statistics section, you have stated “To evaluate differences in BMI between groups, one-way ANOVA tests were conducted. Post-hoc analysis was performed to examine significant interactions between variables, including sports level and gender.” The post-hoc test appears to be on a higher-level ANOVA. I would suggest a 2-factor ANOVA with sport level and sex as factors.
Line 225: “carved out significant differences according to sports level by sports level” – this needs re-wording.
Comments on the Quality of English Language
The English only requires very minor editing
Reviewer 4 Report
Comments and Suggestions for Authors
Dear Authors,
Thank you for submitting your manuscript. Please find my comments below:
In the Abstract, it would be helpful to clarify that non-athletes were included in the study as a control group (line 254). Additionally, please provide the mean age and age range of the study participants in the abstract.
In Section 2.1, indicate the average time required to complete the survey.
Table 1 appears to be redundant and could be summarized in the text. Were most of the study participants within the normal BMI range? The purpose of Table 2 is unclear. What do the presented norms represent? If I understand correctly, all participants were adults aged 18 years and older. The mean age and age range for all groups should be provided in Section 2.2.
Furthermore, in the Statistical Analysis section, several new variables are introduced, such as social media use, body image, and body satisfaction, which are not described in the Methods section. Please clarify these variables.
In the Results section, some sociodemographic data, such as education, are mentioned but not previously introduced in the Methods section. Additionally, the variable "nutritional knowledge sources" presented in Table 4 is not clearly explained and is not presented in the Methods section. How is this variable related to the study's objectives?
For all tables, I recommend either providing exact p-values or marking significant differences with an asterisk (*) and explaining its meaning in the table footnotes.
The presentation of Tables 4 to 7 is incorrect, and the application of the Chi-square test is problematic. The main assumption of the Chi-square test has been overlooked: if more than 20% of the cells in your contingency table have expected frequencies less than 5, the test may not be reliable, increasing the risk of Type I or Type II errors. If any cell has an expected frequency of zero, the Chi-square test cannot be performed, as it would involve division by zero in the formula.
The meaning and presentation of Figures 1 and 2 are unclear. This information could be better integrated into the sample characteristics or participant descriptions.
Overall, the Methods and Results sections, along with the data presentation in tables and figures, are confusing and disorganized.
Please ensure consistent terminology throughout the text—using either "gender" or "sex" and "men" or "males," "women" or "females"—to avoid confusion.
The Discussion and Conclusion sections should be revised after substantial corrections to the Methods and Results sections.
Comments on the Quality of English LanguageNeeds extensive English editing.